# Epidemic Intelligence Service Alumni in Public Health Leadership Roles

**DOI:** 10.3390/ijerph19116662

**Published:** 2022-05-30

**Authors:** Marvin So, Andrea Winquist, Shelby Fisher, Danice Eaton, Dianna Carroll, Patricia Simone, Eric Pevzner, Wences Arvelo

**Affiliations:** 1Oak Ridge Institute for Science and Education, Oak Ridge, TN 37830, USA; 2Epidemiology Workforce Branch, Centers for Disease Control and Prevention, Atlanta, GA 30329, USA; aiw1@cdc.gov (A.W.); dhe0@cdc.gov (D.E.); feu9@cdc.gov (D.C.); pms6@cdc.gov (P.S.); ecp9@cdc.gov (E.P.); dwi4@cdc.gov (W.A.); 3Williams College, Williamstown, MA 01267, USA; shelbyfisher10@gmail.com; 4Commissioned Corps, U.S. Public Health Service, Rockville, MD 20852, USA

**Keywords:** workforce, leadership, applied epidemiology, Epidemic Intelligence Service, Centers for Disease Control and Prevention

## Abstract

Since 1951, the Epidemic Intelligence Service (EIS) of the U.S. Centers for Disease Control and Prevention (CDC) has trained physicians, nurses, scientists, veterinarians, and other allied health professionals in applied epidemiology. To understand the program’s effect on graduates’ leadership outcomes, we examined the EIS alumni representation in five select leadership positions. These positions were staffed by 353 individuals, of which 185 (52%) were EIS alumni. Among 12 CDC directors, four (33%) were EIS alumni. EIS alumni accounted for 29 (58%) of the 50 CDC center directors, 61 (35%) of the 175 state epidemiologists, 27 (56%) of the 48 Field Epidemiology Training Program resident advisors, and 70 (90%) of the 78 Career Epidemiology Field Officers. Of the 185 EIS alumni in leadership positions, 136 (74%) were physicians, 22 (12%) were scientists, 21 (11%) were veterinarians, 6 (3%) were nurses, and 94 (51%) were assigned to a state or local health department. Among the 61 EIS alumni who served as state epidemiologists, 40 (66%) of them were assigned to a state or local health department during EIS. Our evaluation suggests that epidemiology training programs can serve as a vital resource for the public health workforce, particularly given the capacity strains brought to light by the COVID-19 pandemic.

## 1. Introduction

In 1951, the U.S. Centers for Disease Control and Prevention (CDC) inaugurated the first class of its foundational fellowship in field and applied epidemiology, known as the Epidemic Intelligence Service (EIS). One of the primary objectives of the EIS program was to develop a cadre of epidemiologists to bolster the applied epidemiology workforce capacity of federal, state, and local government public health agencies, both domestically and internationally [1,2]. The long-term goal of the fellowship is for these epidemiologists to serve in leadership roles in public health settings. Initially, the EIS focused on training physicians in the methods and strategies of field and applied epidemiology, and disease surveillance. Over the years, the EIS was expanded to include doctoral-level scientists in disciplines related to public health (e.g., epidemiology, social and behavioral sciences), nurses, veterinarians, and some allied healthcare professionals [3].

The EIS is a two-year applied fellowship that fosters the development of epidemiological skills. The fellowship curriculum, which was described in-depth elsewhere [2], begins with a training course over approximately four weeks that introduces the EIS officers to the principles of applied epidemiology including public health surveillance, investigation of outbreaks, health communication, and emergency response. After the EIS officers obtain a foundation of basic epidemiological knowledge, they report to their host sites at CDC, or at a state, local, or territorial health department where they apply these concepts under the guidance of experienced supervisors, many of whom are EIS alumni [2]. In their host sites, officers gain practical skills through hands-on learning experiences, while simultaneously providing essential public health services and serving as frontline responders in public health emergencies. During the EIS fellowship, each officer is required to participate in a field investigation, evaluate a public health surveillance system, analyze public health data, and present their own scientific work in the form of manuscripts, abstracts, reports, and oral presentations [4]. The emphasis on practical experience gives the EIS officers the opportunity to develop the skills necessary to become epidemiologists who are equipped to address a spectrum of public health challenges. Among the EIS graduates in recent classes, >85% have worked in public health-related fields immediately after fellowship completion [2,5].

This evaluation is guided by a framework, which describes the association between program and training activities and intended outcomes. One long-term goal of the program is that trained alumni serve in leadership roles in public health settings. Although a commentary by Thacker et al. in 2001 identified that 43% of state epidemiologists and 3 out of 10 (30%) CDC directors were EIS alumni [2], no systematic evaluations were conducted to characterize the extent to which EIS alumni have served in these and other essential leadership positions. In this evaluation, we seek to provide evidence for a fundamental evaluation question: “Do alumni from epidemiology training programs serve as leaders within the public health workforce?” To answer this question, we examined the number of the EIS alumni among the individuals who have served in selected public health workforce leadership roles at CDC and state or local health departments. Among the EIS alumni who assumed the selected leadership positions, we also examined their characteristics and their EIS fellowship experience. We evaluated success towards this goal in an attempt to inform how epidemiology training programs might serve the long-term goal of funneling leaders into the public health workforce, particularly given the current COVID-19 pandemic.

## 2. Materials and Methods

We focused in this case study on five essential leadership positions at CDC or state or local health departments: CDC director; CDC center director; state epidemiologist; Field Epidemiology Training Program (FETP) resident advisor; and Career Epidemiology Field Officers (CEFOs). Though there are many other positions of leadership at CDC, we selected these five essential leadership positions based on the more readily available data sources [6]. These selected positions represent varying degrees of leadership, and all of these positions were also included because the structure and demands of the jobs require unique skills of critical thinking and evidence-based decision-making, particularly around surveillance, epidemiology, and statistical analysis, that the EIS seeks to instill in its graduates to address public health needs at local, territorial, state, national, and international levels.

CDC directors lead the largest public health organization in the United States, with the main objectives of improving national and global health security, reducing the largest causes of morbidity and mortality, and supporting and strengthening the collaboration between public health and healthcare systems [7]. CDC center directors organize and oversee the work of their respective centers within the CDC [8]. Although other critical leadership positions at the division, branch, and team levels of CDC exist, the historical records are not easily traceable due to the agency’s many reorganizations. State epidemiologists oversee state- or territory-wide programs for the surveillance, prevention, and control of diseases, coordinate epidemiologic investigations and surveys, and manage intervention services [9]. They also coordinate disease surveillance and research activities of federal, state, and local officials in their jurisdictions, and communicate with media sources and public officials through conferences and policy briefs. FETP resident advisors are epidemiologists placed in countries around the world to provide technical and epidemiological assistance to ministries of health. Resident advisors are responsible for planning and organizing the implementation of FETPs in their host countries in a manner that best supports the epidemiologic needs of the host country or region, as well as supervising and mentoring their trainees [10,11]. CEFOs are federally funded CDC epidemiologists placed in state and major metropolitan health departments to strengthen their health departments’ emergency preparedness and capacity for effective public health response [12].

First, we constructed a dataset with the names of people who have held each of the five selected positions. For several reasons, the evaluation focused on positions held from 2000–2016 for all positions except for the CDC Director. First, the CEFO program was established in 2001 following the attacks of 11 September 2001. Second, data were readily available through this period for all five positions. Finally, for the position of CDC director, we were able to obtain data dating back to 1953, so this position was evaluated across a longer time period than the others.

Using the History section of the online CDC webpage, “About CDC 24-7” (https://www.cdc.gov/about/history/pastdirectors.htm, accessed on 25 May 2018), we found the tenure dates and names of all of the CDC directors and acting directors who served the agency during 1953–2016 (the first four directors who served during 1942–1953 were not included because they started their tenure before the first EIS class finished the fellowship). To compile the names of the CDC center directors, one investigator directly inquired with points of contact for each CDC center and the National Institute for Occupational Safety and Health and requested the names and tenure dates of every director during 2000–2016. We also reviewed past CDC organizational charts, obtained using the Internet Archive (San Francisco, CA, USA, https://archive.org/web/, accessed on 25 May 2018). For this evaluation, we defined the CDC center directors as the people listed by name in CDC’s organizational chart during their respective term of service, including the directors of CDC coordinating centers, institutes, and offices that were not horizontally related to the office of the director in the organizational chart [13]. To compile the names of state epidemiologists who served during 2002–2016, we used information posted on the Council of State and Territorial Epidemiologists (CSTE) web site, using the Internet Archive. The Internet Archive had copies of CSTE’s web site dating back to 2002. To compile the list of state epidemiologists who served during 2000–2001, we used appendices of MMWR Surveillance Summaries (https://www.cdc.gov/mmwr/indss_2018.html, accessed on 25 May 2018), which listed state epidemiologists for those years. We obtained a list of names and tenure dates of Resident Advisors who served during 2000–2016 from an administrative database maintained by FETP at CDC’s Center for Global Health. We acquired names and service dates of all CEFOs serving during 2001 (the year in which the CEFO program began) through 2016 from the CDC’s Center for Preparedness and Response, which oversees CEFOs. For all types of positions, those designated as acting or interim were not included (i.e., people serving in the role temporarily in between terms of officially hired or appointed people).

We used an internal electronic dataset of all of the individuals who participated in the EIS fellowship since 1951, stored in CDC’s Fellowship Management System. Data elements included the first and last name of each officer, fellowship start year, host site assignment, and professional category (physician, nurse, scientist, nurse, or veterinarian). Scientists included individuals with PhDs or similar terminal doctoral degrees, such as a DrPH. Professional categories were assessed in the following order which prioritized an individual’s clinical degree (with the finalized assignment being the first category that characterized the officer): physician; veterinarian; nurse; or doctoral scientist. For example, a physician who also had a scientific doctoral-level degree would be considered to be a physician.

To match the names of those identified as serving in the selected leadership positions with the names of the EIS alumni, we first used a probabilistic matching program [14] in SAS^®^ version 9.1 (SAS Institute, Cary, NC, USA). The program compared the first and last names in the EIS’s alumni list with the first and last names in the lists of the individuals serving in each leadership position, and calculated the complex agreement patterns that allow for possible typographical errors. The edit distance proportion was used to develop match scores indicating the likelihood that the two records were a match. Records with match scores > 0 were categorized as possible matches. Possible matches were reviewed manually. Finally, unmatched records in lists of the individuals serving in leadership positions were also reviewed by the authors and other EIS program staff to identify known matches that were not identified by the matching program due to last name changes or differences in name conventions (e.g., Bill vs. William). We characterized the EIS alumni who had served in selected leadership positions by professional category, type of EIS assignment (CDC vs. state or local health department), and geographic location of the leadership position (for state epidemiologists, resident advisors, and CEFOs). In rare instances when the EIS officer changed assignment type during the fellowship, the assignment type at the end of the fellowship was used for purpose of this analysis. Several people in the sample served multiple times in the same type of leadership position (e.g., served as CEFO in multiple states); each person was counted only once for each type of leadership position. If an individual served in more than one type of leadership position, they were counted for each category they served in. This project was reviewed by the CDC and was deemed non-research, as it was considered a program evaluation that included only retrospective analysis of existing data.

## 3. Results

During the period of our evaluation, 353 individuals occupied the five leadership positions. We identified 12 people who had served as CDC directors during 1953–2016; during our evaluation period of 2000–2016, we identified 50 people who had served as center directors, 175 who had served as state epidemiologists, 48 who had served as FETP resident advisors, and 78 who had served as CEFOs. The people serving in these positions included 353 unique individuals, of which 185 (52%) were identified as EIS alumni. Of these 353 people, 10 had served in >1 type of leadership position during the time frames considered, of which six were EIS alumni.

Among 12 CDC directors during 1953–2016, four (33%) were EIS alumni; collectively these alumni led the agency for approximately 25 years. During 2000–2016, 29 (58%) of the 50 CDC center directors were EIS alumni. State epidemiologists represented the largest group of leaders in our evaluation, 61 (35%) of the 175 were EIS alumni. Of the FETP resident advisors, 27 (56%) of the 48 were EIS alumni, and 70 (90%) of the 78 CEFOs were EIS alumni (Table 1).

Of the 184 EIS alumni serving in leadership positions, almost three-quarters (136 [74%]) were physicians, 22 (12%) were doctoral-level scientists, 21 (11%) were veterinarians, and 6 (3%) were nurses. Physicians were the most frequently represented professional category among the EIS alumni who served in each of the leadership roles examined. All CDC directors who were the EIS alumni were physicians; 97% of the EIS alumni who served as CDC center directors, 87% who served as state epidemiologists, 78% who served as FETP resident advisors, and 51% who served as CEFOs were physicians. Physicians were represented among all five leadership roles; doctoral scientists and veterinarians were represented among people serving in four and three of the roles assessed, respectively. Nurses were represented among State Epidemiologists and CEFOs (Table 1).

Ninety-three (51%) of the EIS alumni holding one of the leadership positions were placed in a field site (i.e., state or local health department), and 91 (49%) were placed during their EIS fellowship at a center within the CDC. Among the four CDC directors who were EIS alumni, two (50%) had their EIS assignment in CDC centers and two (50%) in a field assignment (50%). CDC center directors were more often placed at a CDC center (76%) during their EIS fellowship, as were the FETP resident advisors (70%). Conversely, state epidemiologists and CEFOs had more representation from the EIS alumni who had field assignments (66% and 59% of the EIS alumni in these positions, respectively).

EIS graduates have served as State Epidemiologists and CEFOs throughout the United States, territories, and the District of Columbia during 2000–2016. During this period, 35 states, Puerto Rico, and the U.S. Virgin Islands had a state or territorial epidemiologist who was an EIS alumnus. Thirty-two states, Washington, D.C., Puerto Rico, and the U.S.-affiliated Pacific Islands, had a CEFO who was an EIS alumnus. Twenty-two states and Puerto Rico both had at least one state epidemiologist and at least one CEFO who were EIS alumni during the evaluation period (Figure 1). All but five states have had either a state or territorial epidemiologist or a CEFO who was an EIS alumnus during the evaluation period. FETP resident advisors have served in a total of 29 countries and three regional positions comprising more than one country during the evaluation period. FETP resident advisors who were EIS alumni have served in 19 countries (66% of countries that have resident advisors), and one regional position (Figure 2).

## 4. Discussion

EIS alumni are well-represented among the people serving in select public health leadership positions at national, state, and territorial levels, highlighting the importance of epidemiology training programs in augmenting the public health workforce. In this evaluation, we found that the EIS alumni accounted for between approximately one-third (CDC directors and state epidemiologists) and the majority (up to 90% for CEFOs) of certain leadership positions across the CDC, and state and local health departments. Our evaluation echoes similar findings on the career trajectories of the EIS alumni documented in other reports and commentaries [1,2,3].

Not surprisingly, we found that the majority of the EIS graduates serving as state epidemiologists had a field assignment at a state or local health department during their fellowship. An analysis of the EIS officers serving during 1991–1996 found that officers placed in host sites at state or local health departments were more likely to select a state or local health department as their first post fellowship position, compared with individuals placed at a CDC center [15]. This finding reinforces the importance of having EIS officers assigned to state or local health departments to promote a pipeline of field-based public health leaders trained in applied epidemiology.

We also found that physicians were well-represented among EIS alumni who served in these leadership positions. Many of the alumni in the included leadership positions in this evaluation were from earlier EIS classes, and this might explain the higher percentage of physicians in leadership positions found in our evaluation [16]. In addition, physicians were historically preferentially selected for public health leadership positions overall. Despite this trend, more recent EIS classes have had higher representation of nonphysician professional categories (the 2020 EIS class includes 45% doctoral scientists, 35% physicians, 10% veterinarians, 8% nurses, and a physical therapist) [4]. The initial EIS classes were comprised entirely of physicians, but the program has become more diverse over time to be more inclusive of the breadth of disciplines and experience needed for the public health workforce. For example, officers in recent EIS classes include those with advanced degrees in transportation systems’ engineering, nutrition, translational biomedical sciences, anthropology, and numerous sub-disciplines of epidemiology. If the EIS is a pipeline to leadership irrespective of professional category, we expect an increasing number of EIS trained non-physicians to assume these leadership positions more prominently in the future.

Multiple limitations are worth considering in interpreting the evaluation findings. We did not have longitudinal information about the career trajectories of the EIS alumni, so we could not examine the percentage of the EIS alumni who have served in any leadership positions. Additionally, we focused solely on applied epidemiology leadership positions for which we had access to reliable data. Therefore, our evaluation does not allow us to understand the extent to which the EIS is a pipeline to any leadership position. Our approach effectively excluded other roles outside of applied epidemiology that could conceivably be considered as leadership positions (e.g., health department commissioners), positions in academia (e.g., Dean or faculty positions in schools of public health), or other positions at the state and local level that are crucial for epidemiologic field investigations or emergency response [17]. Relatedly, we were unable to examine additional sociodemographic characteristics of the EIS officers as these data were not consistently collected until recent years; future analyses can help clarify opportunities for expanding the diversity of the EIS with respect to race/ethnicity, gender identity, and other factors. Finally, it should be noted that the selected types of positions differ greatly in the degree to which they represent senior leadership positions. CDC directors and CDC center directors representing relatively more senior positions and CEFOs and FETP resident advisors representing less senior positions. Collectively, our approach leveraged existing data sources to optimize the feasibility of this evaluation and yield timely, actionable insights to guide program improvement. This evaluation is a preliminary step in providing data on the extent to which the EIS is a pipeline to leadership positions at the CDC and at state and local health departments.

In sum, the EIS alumni appear to serve as a critical pipeline for public health leaders, both domestically and internationally. Similarly, we might expect that other epidemiology training programs would also be critical inputs into the public health workforce pipeline. Ideally, such training programs would prepare their graduates for critical leadership roles within the public health workforce. Previous studies have shown that cross-cutting skills that transcend epidemiologic statistical methods and include collaborative engagement, change management, persuasive communication, and systems thinking are among the most important knowledge, skills, and attitudes for the contemporary public health workforce [18,19,20]. Collectively, this literature suggests that public health problems require epidemiologists to possess skillsets beyond research and analytic responsibilities. Ongoing efforts to inculcate these competencies into epidemiology training programs and evaluate their effects on leadership outcomes, remain important.

Supporting epidemiology training programs will be critical in ensuring that a robust pipeline of leaders will be available for the public health workforce, particularly in light of the strains and demands placed on health systems during the COVID-19 pandemic. Additional efforts can be undertaken to advance the capacities of graduates from epidemiology training programs, including a focus on communication science, data science initiatives, and addressing racism as a public health issue. The CDC’s Preventive Medicine Residency and Fellowship seeks to prepare participants for leadership roles through enhancing skills in program evaluation, management, policy, and population health improvement for EIS graduates [21]. In addition, the CDC started the Future Leaders in Infections and Global Health Threats (FLIGHT) program in 2019, to foster leadership development among graduating EIS officers with medical or veterinary degrees. Top performing graduating EIS officers who apply are chosen through a rigorous selection process. The 3-year program is currently designed to provide physicians and veterinarians with advanced training and experiences to build scientific leadership skills, and the capability to translate public health research into effective infectious disease programs with global applications. As the evidence for the value of the leadership training provided by FLIGHT becomes available, this leadership program could be expanded to other non-clinical professionals. Tracking the process and outcome indicators of training initiatives are important for identifying specific workforce development strategies to disseminate at scale.

## 5. Conclusions

Drawing on the data from 353 individuals, we determined that alumni of the CDC’s EIS fellowship are represented among five select leadership positions at multiple levels of the public health workforce. The majority of the EIS alumni that subsequently became state epidemiologists were assigned to a state or local health department during their fellowship, underscoring the importance of placement site in training initiatives to expand the scale of the public health workforce. Evaluations leveraging existing historical records can assist public health workforce development programs in understanding the scope and impact of their efforts. These data can help inform actions to better align training program activities with intended outcomes as we continue to grapple with emergent public health threats.

## Figures and Tables

**Figure 1 ijerph-19-06662-f001:**
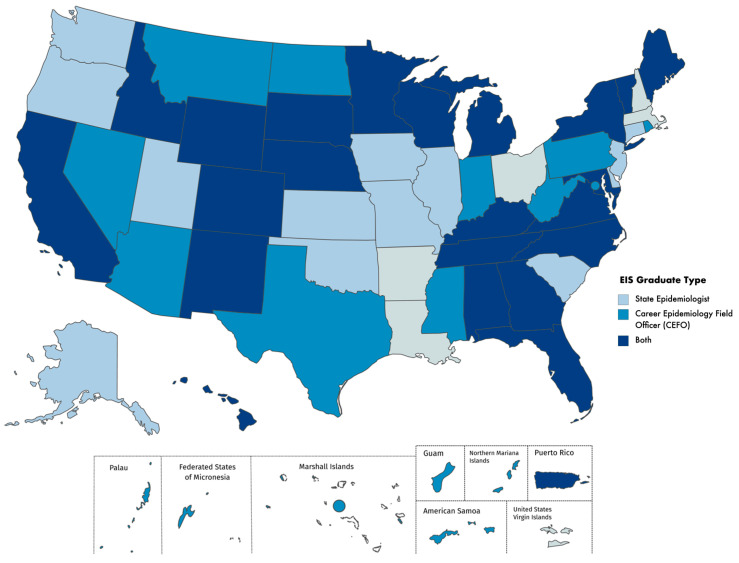
States in which an Epidemic Intelligence Service (EIS) Alumnus Has Served as the State Epidemiologist, Career Epidemiology Field Officer (CEFO), or both positions, 2000–2016. Note: CEFOs assigned to New York were all assigned to the New York City Department of Health and Mental Hygiene. All six U.S.-affiliated Pacific Islands are served by one CEFO.

**Figure 2 ijerph-19-06662-f002:**
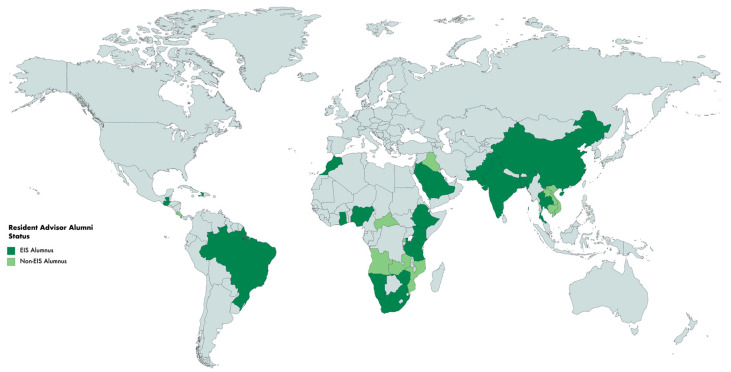
Countries with Field Epidemiology Training Program Resident Advisors, by Epidemic Intelligence Service (EIS) Alumni Status, 2000–2016. Note: Three Career Epidemiology Field Officers were assigned at the regional level and not reflected on this map.

**Table 1 ijerph-19-06662-t001:** Characteristics of Epidemic Intelligence Service (EIS) alumni in select public health leadership positions.

Characteristic	CDC Director	CDC Center Director	State Epidemiologist	FETP Resident Advisor	CEFO	Total, Excluding Duplicates
Years included	1953–2016	2000–2016	2000–2016	2000–2016	2001–2016	N/A
Total number of people serving in position during the included years	12	50	175	48	78	353
Number of EIS alumni serving in position, *n* (%)	4 (33)	29 (58)	61 (35)	27 (56)	70 (90)	185 (52)
Profession, *n* (%)						
Physician	4 (100)	28 (97)	52 (85)	21 (78)	36 (51)	136 (74)
Doctoral scientist	0 (0)	1 (3)	4 (7)	4 (15)	14 (20)	22 (12)
Veterinarian	0 (0)	0 (0)	4 (7)	2 (7)	15 (21)	21 (11)
Nurse	0 (0)	0 (0)	1 (2)	0 (0)	5 (7)	6 (3)
EIS Assignment Type, *n* (%)						
CDC Center or Headquarters	2 (50)	22 (76)	21 (34)	19 (70)	29 (41)	91 (49)
State, territorial, or local health department	2 (50)	7 (24)	40 (66)	8 (30)	41 (59)	94 (51)

Abbreviations: CDC, Centers for Disease Control and Prevention; FETP, Field Epidemiology Training Program; CEFO, Career Epidemiology Field Officer.

## Data Availability

The data presented in this study are available on request from the corresponding author. The data are not publicly available as they are used for internal evaluations and quality improvement activities.

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
