# Peer review of "Epidemic Intelligence Service Alumni in Public Health Leadership Roles"

_ijerph, 2022, doi:10.3390/ijerph19116662_

Round 1

Reviewer 1 Report

Many thanks for the opportunity to read an interesting article on the U.S. Epidemic Intelligence Service. It was with great curiosity that I read about a training program that is now over 70 years old and continues to provide professional staff in epidemiology for CDC.

In my opinion the title does not reflect the content of the paper, so suggests changing the title. The reference in the title of the article to the COVID-19 pandemic is, in my opinion, definitely over the top.

I very much appreciate the effort put into the analysis. Nevertheless, the quantitative analyses carried out are of limited value when it comes to the evaluation of EIS training, as the authors are well aware. Moreover, the authors themselves did not use the potential of quantitative analysis by omitting aspects such as gender or ethnicity of the graduates, which could have been potentially scientific significant.

It seems that the title does not reflect the content of the paper, so suggests changing the title. The reference in the title of the article to the COVID-19 pandemic is, in my opinion, definitely over the top.

Author Response

Response: Thank you for this review. We agree that the title should be changed to more accurately reference what was done and have done so.

We understand the reviewer’s concern about the quantitative analyses. As we state in the manuscript, the goal of this evaluation was primarily to describe the extent to which EIS alumni are represented in select public health leadership roles as this has not been previously done; this was also mentioned as a limitation in the Discussion section. Unfortunately, gender and ethnicity of graduates was not routinely collected since the inception of EIS, thereby precluding our ability to examine these variables. This would certainly represent a valuable area for future workforce research.

Reviewer 2 Report

Thank you for the opportunity to review this interesting manuscript on developing public health leaders for the post-pandemic era. The authors conclude that that epidemiology training programs can serve as a vital resource for the public health workforce. The results are presented clearly and the limitations are discussed appropriately. References are fine (please provide publishing source and date of ref 21). I have no further comments and would recommend publication without major revisions. Best regards and good luck.

Author Response

Response: Thank you for this reviewer’s comments and note about reference #21. We have added the publishing and access date for that reference.

Reviewer 3 Report

In their paper, the authors assessed whether public health leaders who held five different kind of leadership positions in the public help domain (mostly between 2000 and 2016) were Epidemic Intelligence Service alumni i.e., whether they had followed a public health training curriculum emphasizing on practical experience.

The article is interesting, well written, and addresses an important question, all the more so in light of the COVID-19 pandemic. Indeed, I believe that public health leaders should be adequately trained, and it is crucial to determine whether trained professionals finally choose to (or are able to) work in the field they were trained for. The limitations of this study are acknowledged and perspectives are drawn.

Major comments: none.

Minor comments
1. Line 52: There is an issue with reference number 4 (not between brackets and after punctuation, just a word-processor / reference manager issue).
2. Line 76: FETP should be defined (currently first defined in Table 1, line 184)
3. Line 77: CEFOs should be defined (currently first defined in Table 1, line 184)
4. Lines 89-91: "There are many other highly 89 important leadership positions at the CDC at the division, branch, and even team and 90 levels." - the last "and" should be removed

Author Response

Response: Thank you for this review and the reviewer’s commendation and helpful points. With respect to minor comments, the Line 52 bracket issue has been fixed; FETP and CEFOs have been defined at its first use in-text; and we have removed the additional “and” in Lines 89-91.

Reviewer 4 Report

The article presented for review contains an interesting case report based on the assessment of quality and effectiveness of education in Epidemic Intelligence Service (EIS) fellowship. The conducted analysis, based on historical sources, was intended to answer the main research question of the study: “Do alumni from epidemiology training programs serve as leaders within the public health workforce?”.

The study was clearly described, methodologically correct, and finally carried out carefully. The results are presented in a descriptive form, as well as illustrated with graphics and a table.

I have three, minor comments: 

A curriculum and teaching staff could be described more detailed.

The limitation paragraph should be added. For example, as the authors pointed out, the analysis of people who have held each of the  five selected positions concerns not the entire history of education, but rather the period 2001-2016. Unfortunately, this fact does not make it possible to assess the functioning of graduates in the period of probably the greatest trial, which was (and perhaps still is) the COVID-19 pandemic.

Author Response

Response: Thank you for this reviewer’s commendations. We appreciate the suggestion about adding more information about the EIS curriculum and teaching staff; however, the curriculum has been described in-depth elsewhere thus we point readers to reference #2 (Line 42). Regarding the suggestion about Limitations, a Limitations paragraph is already present (Line 273). As we note in our response to Reviewer #1, we have de-emphasized the focus on COVID-19 in the paper’s title. We agree that the current evaluation does not permit us to understand how EIS officers have performed in the context of COVID-19, though this is an area of ongoing data collection by our team.